# RECESS Vaccine for Federated Learning: Proactive Defense Against Model Poisoning Attacks

**Haonan Yan[1,2], Wenjing Zhang[2], Qian Chen[1], Xiaoguang Li[1]***
**Wenhai Sun[3], Hui Li[1]*, Xiaodong Lin[2]**
[1]Xidian University, [2]University of Guelph, [3]Purdue University
`yanhaonan.sec@gmail.com`

## Abstract

Model poisoning attacks greatly jeopardize the application of federated learning (FL). The effectiveness of existing defenses is susceptible to the latest model poisoning attacks, leading to a decrease in prediction accuracy. Besides, these defenses are intractable to distinguish benign outliers from malicious gradients, which further compromises the model generalization. In this work, we propose a novel defense including detection and aggregation, named RECESS, to serve as a "vaccine" for FL against model poisoning attacks. Different from the passive analysis in previous defenses, RECESS proactively queries each participating client with a delicately constructed aggregation gradient, accompanied by the detection of malicious clients according to their responses with higher accuracy. Further, RECESS adopts a newly proposed trust scoring based mechanism to robustly aggregate gradients. Rather than previous methods of scoring in each iteration, RECESS takes into account the correlation of clients' performance over multiple iterations to estimate the trust score, bringing in a significant increase in detection fault tolerance. Finally, we extensively evaluate RECESS on typical model architectures and four datasets under various settings including white/black-box, cross-silo/device FL, etc. Experimental results show the superiority of RECESS in terms of reducing accuracy loss caused by the latest model poisoning attacks over five classic and two state-of-the-art defenses.

## 1 Introduction

**Background and Problem.** Recently, federated learning (FL) goes viral as a privacy-preserving training solution with the distributed learning paradigm [1], since data privacy attracts increasing attention from organizations like banks [2] and hospitals [3], governments like GDPR [4] and CPPA [5], and commercial companies like Google [6]. FL allows data owners to collaboratively train models under the coordination of a central server for better prediction performance by sharing local gradient updates instead of their own private/proprietary datasets, preserving the privacy of each participant's raw data. FL is promising as a trending privacy training technology. However, FL is vulnerable to various model poisoning attacks [7, 8]. The distributed structure of FL characterized by the privacy-preserving local training paradigm makes it impossible to verify overall gradient updates and distributions of local datasets. Accordingly, the attacker can corrupt the global model by hijacking compromised participants, and then uploading malicious local gradient updates, leading to a reduction in the final model's prediction performance, which significantly jeopardizes the application of FL.

---

*Corresponding author

37th Conference on Neural Information Processing Systems (NeurIPS 2023).

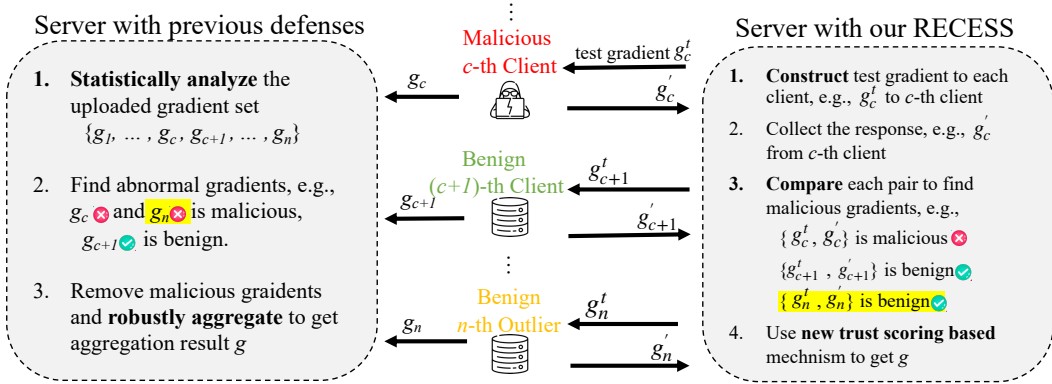

Figure 1: An intuitive example to illustrate the difference between previous defenses and RECESS. In an FL system with $n$ clients, suppose the first $c$ clients are malicious. Previous defenses find the malicious gradients by abnormality detection of all received gradients, while RECESS confirms the malicious gradients by comparing the constructed test gradient with the corresponding client's response. Thus, RECESS can distinguish between malicious clients and benign outliers, achieving higher accuracy in the final trained model.

**Limitations of Previous Works.** To mitigate model poisoning attacks, many byzantine-robust aggregation rules, e.g., Krum, Mkrum, Trmean, Median, and Bulyan [9, 10, 7], are proposed to remove malicious gradients by statistical analyses. Two state-of-the-art (SOTA) defenses, FLTrust [11] and DnC [12], are also proposed to further enhance the robustness of the FL system. Even so, (**Limitation 1:**) *these defenses are susceptible to the latest optimization-based model poisoning attacks [13, 12], leading to a decrease in prediction accuracy.*

Besides, (**Limitation 2:**) *these defenses cannot distinguish malicious gradients and benign outliers, reducing the generalization performance of the final trained model.* Considering that the local datasets across participating clients are Non-IID, it is common to have many biased updates which are statistical outliers. These statistical outliers are still benign updates (called *benign outliers*), which are helpful in improving the generalization performance of the final FL model. However, existing defenses typically rely on statistical majorities to remove malicious clients, which also implicates these benign outliers. To the best of our knowledge, the work of tackling benign outliers still remains open in FL.

**Our work.** To improve model accuracy and generalization against latest model poisoning attacks in FL, we propose RECESS acting as "vaccine" to help FL pRoactively dEteCt modEl poiSoning attackS. Unlike previous defenses using passive analysis and direct aggregation in a single iteration, RECESS *proactively detects* malicious clients and *robustly aggregates* gradients with a new trust scoring based mechanism. The differences are illustrated in Figure 1. In *proactive detection*, our key insight is that the update goals of malicious clients are different from benign clients and outliers. Therefore, the defender can amplify this difference by delicately constructing aggregation gradients sent to clients, and then more accurately detect malicious clients and distinguish benign outliers by the comparison between constructed gradients and the corresponding responses. In *robust aggregation*, we propose a new trust scoring based mechanism which estimates the score according to the user's performance over multiple iterations, rather than scoring each iteration as in previous methods, which improves the detection fault tolerance. Finally, we compare RECESS with five classic and two SOTA defenses under various settings, and the results demonstrate that RECESS is effective to overcome the two limitations aforementioned.

**Contribution.** The main contributions are:

(a) We propose a novel defense against model poisoning attacks called RECESS. It offers a new defending angle for FL and turns the defender from passive analysis to proactive detection, which defeats the latest model poisoning attacks. RECESS can also identify benign outliers effectively.

(b) We improve the robust aggregation mechanism. A new trust scoring method is devised by considering clients' abnormality over multiple iterations, which significantly increases the detection fault tolerance of RECESS and further improves the model's accuracy.

(c) We evaluate RECESS under various settings, including white/black-box, Non-IID degree of the dataset, number of malicious clients, and cross-silo/device FL. Experimental results show that RECESS outperforms previous defenses in terms of accuracy loss and achieves consistent effectiveness.

## 2 Related Works

### 2.1 Latest Model Poisoning Attacks in FL

In this work, we focus on the stronger untargeted model poisoning attacks for three reasons:

(a) Model poisoning attacks with the direct manipulation of gradients are more threatening than data poisoning attacks in FL [8, 13].

(b) Data poisoning attacks are considered a special case of model poisoning attacks as malicious gradients can be obtained on poisoning datasets [11].

(c) Untargeted poisoning is a more severe threat to model prediction performance than the targeted one in FL [12].

In the following, we introduce three latest model poisoning attacks.

**LIE Attack.** The LIE attack [14] lets the defender remove the non-byzantine clients and shift the aggregation gradient by carefully crafting byzantine values that deviate from the correct ones as far as possible.

**Optimization Attack.** [13] propose a new attack idea. They formulate the model poisoning attack as an optimization program, where the objective function is to let the poisoning aggregation gradient be far from the aggregation gradient without poisoning. By leveraging the halving search, they obtain a crafted malicious gradient to poison.

**AGR Attack Series.** [12] then improve the optimization program by introducing perturbation vectors and scaling factors. Then three instances (AGR-tailored, AGR-agnostic Min-Max, and Min-Sum) are proposed, which maximize the deviation between benign gradients and malicious gradients.

### 2.2 Existing SOTA Defenses

There are two directions to defend against the model poisoning attack: robust aggregation and anomaly detection. Five classic byzantine-robust aggregation rules and FLTrust belong to the robust aggregation. DnC is one of the representative anomaly detectors. Here we mainly describe two SOTA defenses.

**FLTrust.** [11] devises FLTrust to mitigate the poisoning attack. In FLtrust, the server maintains a small clean dataset and act as a client to participate in the update, adding the deployment constraint. In each iteration, clients' gradients whose direction deviates more from the gradient of the server are discarded. Moreover, FLtrust normalizes the gradient magnitude to limit the impact of malicious gradients and achieves a competitive accuracy.

**DnC.** [12] leverages the spectral method to detect malicious gradients, which is proven effective in centered learning. They also use random sampling to reduce memory and computational cost. Note we will evaluate these two defense mechanisms, which have rigorous theoretical guarantees, as baselines.

## 3 Proposed RECESS

We propose a new proactive defense against model poisoning attacks in FL called RECESS. We first take an overview of RECESS. Then we introduce the proactive detection and the new trust scoring based aggregation mechanism respectively, followed by the effectiveness analysis.

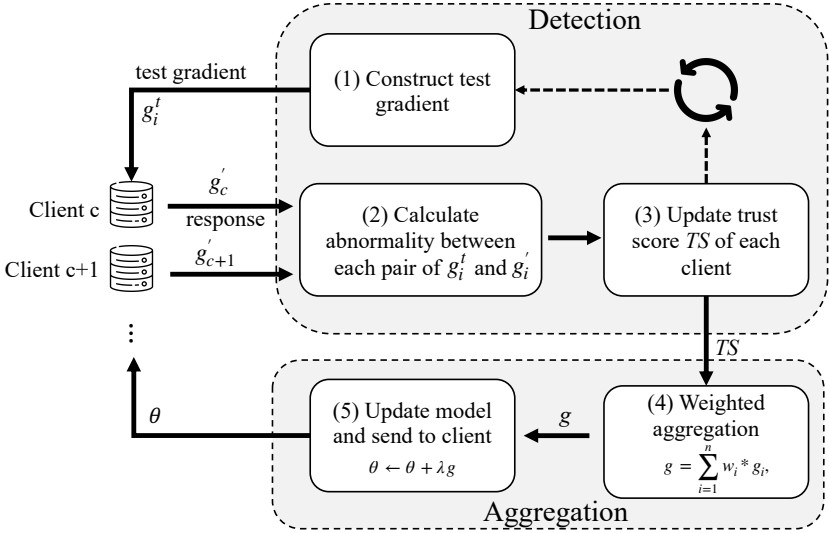

Figure 2: The overall detection process of RECESS against model poisoning attacks.

## 3.1 Overview

**Intuition.** RECESS aims to distinguish malicious and benign gradients without implicating benign outliers. The essential difference between benign and malicious clients is that: benign clients including outliers always optimize the received aggregation result to the distribution of their local dataset to minimize the loss value, while malicious clients aim to make the poisoned aggregation direction as far away from the aggregation gradient without poisoning as possible to maximize the poisoning effect. In other words, benign gradients explicitly point to their local distribution, which is directional and relatively more stable, while the malicious gradients solved based on other benign gradients change inconsistently and dramatically. The workflow of RECESS is shown in Figure 2.

**Proactive Detection.** RECESS is different from previous passive analyses. The defender using RECESS proactively tests each client with elaborately constructed aggregation gradients, rather than the model weights to clients, and these two approaches are equivalent algorithmically. After clients fine-tune local models with received aggregation gradients and respond with new gradient updates, the defender can recognize malicious clients based on the abnormality of their gradient updates. We also redefine the abnormality in poisoning scenarios to distinguish malicious gradients and outliers, which promotes the generalization of the FL model. Details are shown in Section 3.2.

**Robust Aggregation.** RECESS adopts a new trust scoring based aggregation mechanism to improve the defense. A new scoring method is proposed to give weights to aggregate clients' gradients. Updates with greater abnormal extent account for a smaller proportion in aggregation, which increases fault tolerance. Details are shown in Section 3.3.

## 3.2 Proactive Detection Algorithm

**Construction Strategy of Test Gradient.** The purpose of the defender is to observe the client's response by adding a slight perturbation to the test gradient. Thus, any slight varying gradient is feasible. To illustrate, we present a strategy based on previous gradients. In details, the defender firstly stores the aggregation gradient in the last iteration. When entering the detection mode, for example in $(k-1)$-th round of detection, the defender idles this iteration, and for each uploaded gradient $g_i^{k-1}$ from $i$-th client ($0 < i \leq n$, $n$ is the number of clients), the defender scale down the magnitude of $g_i^{k-1}$, i.e., $\|g_i^t\|_2 = g_i^{k-1}/\|g_i^{k-1}\|_2$, and slightly adjust the direction of $g_i^t$ by heuristically selecting several elements and adding noise. Here we set a threshold $\mathcal{A}$ to control the direction adjustment. Then, the defender feedbacks $g_i^t$ as the aggregation gradient to client $i$. The client $i$ updates this tailored "aggregation gradient" locally and respond with a newly updated gradient $g_i^k$. The defender

can perform poisoning detection by the comparison between $\boldsymbol{g}_i^t$ and $\boldsymbol{g}_i^k$. This process will be repeated for each client.

RECESS slightly adjusts constructed aggregation gradients, which magnifies the variance of malicious gradients and makes them more conspicuous, but benign clients and outliers are not affected. The reason is: for benign clients including outliers, their updated gradients always point to the direction of decreasing loss function values. Although some machine learning optimization algorithms, such as SGD, have deviations, they are still unbiased estimations of the normal gradient descent direction as a whole [15]. Conversely, malicious clients' gradients are usually opposite to the aggregation direction and come from the solution of the attack optimization program on other benign gradients. Thus, the variance of malicious gradients is enlarged by the optimization project. Many works [16, 17, 9, 8] also indicate that the fluctuation of the upload gradients cause the aggregation gradient to change more. Even when the defender uses the poisoned aggregation gradient to construct test gradient, this basis still remains unchanged and malicious and benign clients behave quite differently in this test. Besides, RECESS without extra model training is more efficient than other model-based detectors. Concluding, RECESS is effective to detect model poisoning attacks.

**Poisoning Detection.** After receiving clients' responses, the defender detects malicious clients from two dimensions of abnormality: *direction* and *magnitude* of gradient changes before and after the test. Formally, we use metric *cosine similarity* $S_C^k$ to measure the angular change in direction between the constructed gradient $\boldsymbol{g}_i^t$ and the response $\boldsymbol{g}_i^k$,

$$S_C^k(\boldsymbol{g}_i^t, \boldsymbol{g}_i^k) = \frac{\boldsymbol{g}_i^t \cdot \boldsymbol{g}_i^k}{\|\boldsymbol{g}_i^t\|_2 \cdot \|\boldsymbol{g}_i^k\|_2}. \tag{1}$$

Besides direction, the magnitude of the malicious gradient also dominates the poisoning effect, especially when larger than the benign gradients. Here we utilize the $l_2$ distance to measure the magnitude, i.e., $\|\boldsymbol{g}_i^k\|_2$.

After that, we redefine the abnormality of the client gradient by the deviation extent in direction and magnitude, instead of the distance from other selected gradients (e.g. Krum, Mkrum, Median, server's gradient [11]) or some benchmarks (e.g. Trmean, Bylan).

**Definition 1** *(Abnormality)* In model poisoning attacks of FL, the abnormality $\alpha$ of $k$-th uploaded gradient from the $i$-th client should be measured by

$$\alpha = -\frac{S_C^k}{\|\boldsymbol{g}_i^k\|_2}. \tag{2}$$

The cosine similarity $S_C^k$ controls the positive and negative of the abnormality $\alpha$. When the direction of the client's response gradient $\boldsymbol{g}_i^k$ is inconsistent with the defender's test gradient $\boldsymbol{g}_i^t$, $\alpha$ will be positive, and vice versa. The amount of change in $\alpha$ is related to $S_C^k$ and $\|\boldsymbol{g}_i^k\|_2$. The smaller the deviation between the direction of $\boldsymbol{g}_i^t$ and $\boldsymbol{g}_i^k$, the smaller the $\alpha$. Meanwhile, the larger the $\|\boldsymbol{g}_i^k\|_2$, the higher the $\alpha$. This setting encourages small-gradient updates, which avoids the domination of aggregation by malicious updates usually with a larger magnitude.

### 3.3 New Robust Aggregation Mechanism

After detection, we propose a new trust scoring based mechanism to aggregate gradients, which increases RECESS's fault tolerance for false detection.

**Trust Scoring.** RECESS estimates the trust score for the long-term performance of each client, rather than the single iteration considered by previous schemes.

The detail of the scoring process is: the defender first sets the same initial trust score $TS_0$ for each client. After each round of detection, a constant baseline score is deducted for clients who are detected as suspicious poisoners and not selected for aggregation. When the trust score of one client $i$ reaches zero, i.e., $TS_i = 0$, the defender will label this client as malicious and no longer involve this client's updated gradients in the aggregation. The $TS$ is calculated by

$$TS = TS_0 - \alpha * baseline\_decreased\_score. \tag{3}$$

The $TS_0$ and $baseline\_decreased\_score$ controls the detection speed.

To prevent the attacker from increasing the trust score by suspending poisoning or constructing benign gradients, we let the defender defer adding $TS$ to further restrict the attacker. Only after a period of good performance (e.g., 10 consecutive rounds), $TS$ increase, but the score would be deducted once the client is detected as malicious, which effectively forces the attacker not to poison or reduce the intensity of poisoning to a negligible level.

**Aggregation.** After assigning $TS$ to all clients, we transform $TS$ into weights which are used to average clients' gradients as the aggregation gradient $g$, i.e.,

$$g = \sum_{i=1}^{n} w_i * g_i, \tag{4}$$

the weight of client $i$ is calculated by

$$w_i = \frac{e^{TS_i}}{\sum_{j=1}^{n} e^{TS_j}}, \tag{5}$$

where $i, j = 1, 2, ..., n$.

**Advantages.** Compared with previous methods, RECESS has four advantages:

(a) Provide a more accurate measure of the abnormality extent for each client's update to assign the trust score.

(b) Enable the defender to put clients' previous performance into account, not only in each iteration, enhancing the detection fault tolerance.

(c) Protect benign outliers while effectively detecting carefully-constructed poisoning gradient updates, improving model accuracy and generalization.

(d) Outperform previous FLTrust and ML-based detections in terms of efficiency with no need for extra data.

## 4 Evaluation

### 4.1 Setup

**Datasets and FL Setting.** Table 1 shows four datasets and parameter settings used in the evaluation. The IID and Non-IID are both considered in the dataset division. We also consider two typical types of FL, i.e., cross-silo and cross-device.

Table 1: Experiment datasets and FL settings.

| Dataset | Class | Size | Dimension | Model | Clients | Batch Size | Optimizer | Learning Rates | Epochs |
|---------|-------|------|-----------|-------|---------|------------|-----------|----------------|--------|
| MNIST | 10 | 60,000 | $28 \times 28$ | FC ($784 \times 512 \times 10$) | 100 | 100 | Adam | 0.001 | 500 |
| CIFAR-10 | 10 | 60,000 | $32 \times 32$ | Alexnet [18] | 50 | 250 | SGD | 0.5 to 0.05 at 1000th epoch | 1200 |
| Purchase | 100 | 197,324 | 600 | FC ($600 \times 1024 \times 100$) | 80 | 500 | SGD | 0.5 | 1000 |
| FEMNIST | 62 | 671,585 | $28 \times 28$ | LeNet-5 [19] | $60 \subset 3400$ | client's dataset | Adam | 0.001 | 1500 |

Eight defenses are considered in the comparison experiments. Specifically, the common FedAvg is used as the benchmark. Five classic robust rules, Krum, Mkrum, Bulyan, Trmean, and Median, can provide convergence guarantees theoretically. Two SOTA defenses, FLtrust and DnC, are also involved. The parameters of these eight defenses are set to default. For RECESS, we set $\mathcal{A} = 0.95$, $TS_0 = 1$, and $baseline\_decreased\_score = 0.1$ unless otherwise specified.

**Attack Setting.** We consider the five latest poisoning attacks including LIE, optimization attack, AGR-tailored attack, AGR-agnostic Min-Max attack, and Min-Sum attack. We also set two scenarios, i.e., white-box and black-box, where attackers have and does not have knowledge of other benign gradients. We assume 20% malicious clients as the default value unless specified otherwise. For the test gradient construction, we select and add noise to the first 10% of dimensions, iteratively adjusting until cosine similarity before/after meets the threshold.

Table 2: Comparison results of FL accuracy with various defenses against the latest poisoning attacks in both white-box and black-box cases. In the white-box (black-box) case, the attacker has (no) knowledge of other benign gradients. Each result is averaged over multiple repetitions.

| Dataset | Attacker's Knowledge | Attacks | Defences | | | | | | | |
|---|---|---|---|---|---|---|---|---|---|---|
| | | | Krum | Mkrum | Bulyan | Trmean | Median | FLTrust | DnC | RECESS |
| CIFAR-10 0.6605 (FedAvg) | | No Attack | 0.5162 | 0.6494 | 0.6601 | **0.6605** | 0.6582 | 0.6341 | 0.6409 | 0.6554 |
| | White-box | LIE | 0.5058 | 0.5921 | 0.0965 | **0.6436** | 0.6243 | 0.6189 | 0.6047 | 0.6085 |
| | | Optimization attack | 0.3459 | 0.5817 | 0.6118 | 0.5066 | 0.5133 | 0.6075 | 0.5813 | **0.6206** |
| | | AGR-tailored | 0.2246 | 0.2915 | 0.2355 | 0.2276 | 0.2970 | 0.4636 | 0.4614 | **0.6043** |
| | | AGR-agnostic Min-Max | 0.5134 | 0.3096 | 0.3173 | 0.3709 | 0.2723 | 0.5229 | 0.5579 | **0.6202** |
| | | AGR-agnostic Min-Sum | 0.2248 | 0.4461 | 0.2211 | 0.4271 | 0.2765 | 0.5476 | 0.5347 | **0.6406** |
| | Black-box | LIE | 0.3139 | 0.4512 | 0.3648 | 0.6063 | 0.5635 | **0.6319** | 0.6101 | 0.6318 |
| | | Optimization attack | 0.2891 | 0.5673 | 0.2605 | 0.5558 | 0.5452 | 0.6284 | 0.6159 | **0.6390** |
| | | AGR-tailored | 0.2455 | 0.3123 | 0.2349 | 0.3155 | 0.2650 | 0.5534 | 0.6066 | **0.6177** |
| | | AGR-agnostic Min-Max | 0.4078 | 0.3311 | 0.2948 | 0.2773 | 0.2934 | 0.5303 | 0.5771 | **0.6286** |
| | | AGR-agnostic Min-Sum | 0.2242 | 0.3435 | 0.2551 | 0.4381 | 0.3492 | 0.5094 | 0.5676 | **0.6432** |
| FEMNIST 0.8235 (FedAvg) | | No Attack | 0.6863 | 0.8206 | 0.7115 | 0.8217 | 0.7897 | **0.8237** | 0.8201 | 0.8230 |
| | White-box | LIE | 0.2687 | 0.5897 | 0.6514 | **0.8146** | 0.7828 | 0.8081 | 0.8048 | 0.7957 |
| | | Optimization attack | 0.2605 | 0.4981 | 0.6286 | 0.7920 | 0.7556 | 0.7848 | 0.7456 | **0.7983** |
| | | AGR-tailored | 0.5462 | 0.3696 | 0.4917 | 0.6127 | 0.4784 | 0.7615 | 0.3661 | **0.8145** |
| | | AGR-agnostic Min-Max | 0.6001 | 0.0606 | 0.6701 | 0.5717 | 0.5552 | 0.7845 | 0.6916 | **0.8019** |
| | | AGR-agnostic Min-Sum | 0.4750 | 0.2904 | 0.4496 | 0.5982 | 0.6264 | 0.7148 | 0.0694 | **0.7967** |
| | Black-box | LIE | 0.2221 | 0.8153 | 0.7045 | 0.8114 | 0.7151 | **0.8148** | 0.8145 | 0.8048 |
| | | Optimization attack | 0.5998 | 0.7411 | 0.6646 | 0.7893 | 0.6741 | 0.7807 | 0.7824 | **0.7937** |
| | | AGR-tailored | 0.5419 | 0.4009 | 0.4330 | 0.6858 | 0.4527 | 0.7651 | 0.7027 | **0.7745** |
| | | AGR-agnostic Min-Max | 0.3796 | 0.5631 | 0.6332 | 0.6440 | 0.6605 | 0.6848 | 0.7351 | **0.7850** |
| | | AGR-agnostic Min-Sum | 0.6135 | 0.7524 | 0.7052 | 0.6589 | 0.6785 | 0.7848 | 0.7122 | **0.7936** |

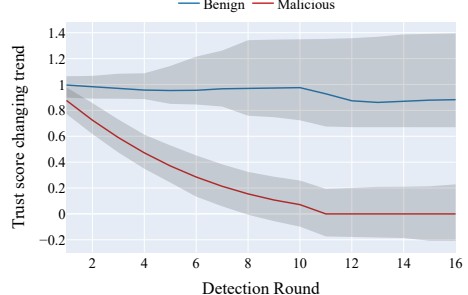

Figure 3: The changing trend of clients' trust score in RECESS. The task is CIFAR-10. The model poisoning attack is the AGR-agnostic Min-Max attack.

Figure 4: The histogram of clients' *Abnormality* value in RECESS. The task is CIFAR-10. The model poisoning attack is the AGR-agnostic Min-Max attack. The detection round is #3.

**Metric.** *Accuracy* is used to measure the performance of the well-trained model with or without poisoning. Higher *Accuracy* indicates better defense.

## 4.2 Comparison with SOTA

Table 2 shows the main comparison results. RECESS obtains higher accuracy than two SOTA defenses under latest poisoning attacks. Due to the space limit, here we mainly show the results on two datasets.

**Defender's Goal (1): Defensive Effectiveness.** As we can see from each row in Table 2, RECESS can defend against model poisoning attacks, but previous defenses have limited effect, especially for the strongest AGR attack series in the white-box case. The reasons are twofold:

(a) Some malicious gradients evade defenses to be selected for aggregation.

(b) Benign outliers are misclassified as malicious gradients and removed from the aggregation, especially for Non-IID datasets (e.g., FEMNIST) with more outliers.

Moreover, benign outliers are intractable for all previous defenses. FLTrust is better than other defenses since its normalization limits the large magnitude gradients. In contrast, RECESS can

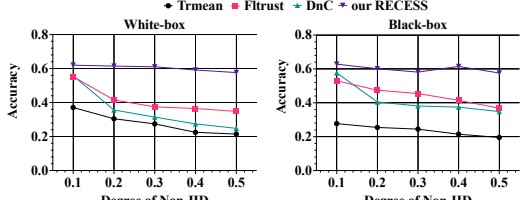
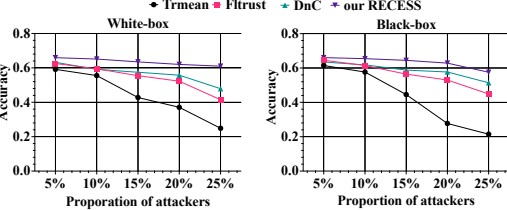

Figure 5: The impact of the local dataset's Non-IID degree on the FL defenses. The task is CIFAR-10. The model poisoning attack is AGR Min-Max. Our RECESS achieves the best and most stable defense effect.

Figure 6: The impact of the proportion of attackers on the FL defenses. The task is CIFAR-10. The model poisoning attack is AGR Min-Max. As the proportion increases, the accuracy of using our RECESS decreases minimally.

effectively distinguish benign outliers from malicious gradients, thus obtaining the highest accuracy, even in the most challenging white-box Non-IID case.

**Defender's Goal (2): Model Performance.** As shown in the first row in each set of Table 2 where no attacks are under consideration, robust rules will reduce the accuracy but have little impact on RECESS. This can be explained from three aspects:

(a) Due to the existence of attackers, only 80% benign clients participate in the aggregation compared with no attack (e.g. 40 vs 50 for CIFAR-10).

(b) Existing defenses discard benign gradients more or less, especially for benign outliers.

(c) Malicious gradients are selected for aggregation.

The first aspect is inevitable since the defender cannot control the attacker, which also causes the main accuracy gap between FedAvg and our RECESS. However, RECESS can improve the other two aspects by more accurately distinguishing whether gradients are malicious or benign.

**Impact of Attacker's Knowledge.** Different knowledge affects the attack, and RECESS outperforms other defenses in both white/black-box. The stronger the attack, the better the effect of RECESS. In the black-box case, the attacker cannot offset the effect of benign gradients on the aggregation without the knowledge of other benign gradients, so the poisoning is greatly weakened and RECESS is not prominent as in the white-box case.

**Changing Trend of Trust Score.** Figure 3 presents the trust score of benign and malicious clients in each round of RECESS detection. Here we execute the strongest AGR-agnostic Min-max attack. Figure 4 shows the histogram of each client's $Abnormality$ value in the third detection round to illustrate the effectiveness of RECESS. RECESS can effectively distinguish malicious and benign users in both the optimization attack and AGR attacks.

### 4.3 Impact of FL Setting

In this part, we evaluate RECESS under various FL settings to illustrate the practicability of our proposed scheme.

**Non-IID Degree of Dataset (More Outliers).** The results on Non-IID FEMNIST in Table 2 show the effectiveness of our RECESS against latest poisoning attacks. Besides, we also involve the CIFAR-10 in the evaluation using the Non-IID dataset construction method same as [11, 12]. We consider four defenses (the best classic robust rules Trmean, SOTA FLtrust and DnC, and RECESS) against the strongest AGR-agnostic Min-Max attack in white/black-box cases.

Figure 5 shows that the increase in the Non-IID degree leads to an expansion of benign outliers which are similar to malicious gradients, and previous defenses cannot effectively distinguish outliers from malicious gradients. However, it has little impact on RECESS, since our method can accurately detect malicious gradients and benign outliers.

Table 3: The impact of cross-device FL on the defenses. The model poisoning attack is AGR Min-Max.

| Dataset | Attacker's Knowledge | Attacks | Defences | | | |
|---|---|---|---|---|---|---|
| | | | Trmean | FLTrust | DnC | RECESS |
| | | No Attack | **0.6490** | 0.6178 | 0.6284 | 0.6448 |
| CIFAR-10 | White-box | LIE | 0.5559 | 0.6101 | 0.6157 | **0.6228** |
| | | Optimization attack | 0.5655 | 0.6171 | 0.5913 | **0.6439** |
| | | AGR-tailored | 0.5080 | 0.4538 | 0.5915 | **0.6046** |
| | | AGR-agnostic Min-Max | 0.5548 | 0.5226 | 0.5478 | **0.6379** |
| | | AGR-agnostic Min-Sum | 0.5758 | 0.5964 | 0.5793 | **0.6393** |
| | Black-box | LIE | 0.5999 | 0.6137 | 0.6128 | **0.6353** |
| | | Optimization attack | 0.6168 | 0.6159 | 0.6209 | **0.6277** |
| | | AGR-tailored | 0.5509 | 0.5485 | 0.5956 | **0.6142** |
| | | AGR-agnostic Min-Max | 0.5833 | 0.5715 | 0.6095 | **0.6416** |
| | | AGR-agnostic Min-Sum | 0.6068 | 0.5937 | 0.6074 | **0.6385** |

Table 4: FL accuracy with enhanced RECESSS defenses against the adaptive poisoning attack. The task is FEMNIST. The detection frequency is contolled by $TS_0$ and $baseline\_decreased\_score$.

| Attacker's Knowledge | Attacks | RECESS Detection Frequency | | | | |
|---|---|---|---|---|---|---|
| | | 0 | 10 | 50 | 100 | 200 |
| | no attack | 0.8235 | 0.823 | 0.8221 | 0.8224 | 0.8215 |
| white-box | optimization attack | 0.2182 | 0.7983 | 0.8048 | 0.8106 | 0.8114 |
| | adaptive optimization attack | 0.2482 | 0.3456 | 0.6847 | 0.7815 | 0.8011 |
| black-box | optimization attack | 0.5152 | 0.7937 | 0.8045 | 0.8117 | 0.8048 |
| | adaptive optimization attack | 0.5248 | 0.6481 | 0.7847 | 0.8048 | 0.8148 |

**Number of Malicious Clients.** Figure 6 shows that RECESS remains an outstanding defending effect all along, as the number of malicious clients increases, all poisoning attacks are more powerful, while the defending effect of other defenses decreases sharply. We vary the proportion of malicious clients from 5% to 25%, consistent with [12]. The other settings remain the same as Section 4.3.

**Cross-device FL.** Previous evaluations are mostly under cross-silo settings except for the FEMNIST which is naturally cross-device shown in Table 2. Further, we consider the cross-device setting using the dataset CIFAR-10. In each epoch, the server stochastically selects 10 updates from all clients to aggregate. The attack steps remain the same, and the other settings are the same as the default.

Table 3 shows that similar to the result of cross-silo, RECESS still achieves the best defense effect. Besides, the poisoning impacts under the cross-device setting are lower than the one of the cross-silo setting, because the server selects less number of clients for aggregation and ignores more malicious gradients in cross-device FL, thus the attacker cannot continuously poison, weakening the impact of the poisoning. Consequently, this leaves less improvement space for RECESS and other defenses.

## 4.4 Adaptive Attacks

**Active Evasion.** With knowledge of RECESS, the attacker checks if it's a test gradient before deciding to poison or not. The adaptive strategy involves estimating an aggregated gradient $g_p$ in each iteration from controlled benign clients. Then, the attacker compares the received aggregated gradient $g_{agg}$ with $g_p$ and $g_i^{k-1}$ from the last iteration. If $S_C(g_{agg}, g_i^{k-1}) \geq S_C(g_{agg}, g_p)$, the poisoning begins, while if $S_C(g_{agg}, g_i^{k-1}) < S_C(g_{agg}, g_p)$, the poisoning stops, as it's considered detected.

As a countermeasure, the defender boosts detection frequency and intersperses it during the entire training, instead of fixed consecutive detection, and increases the sensitivity of parameters. Note that the defense should aim to make the model converge, it is not necessary to find the attacker. In other words, RECESS's presence increases the attack cost, so even if the attacker evades detection, the loss of model accuracy is negligible, which is acceptable.

Table 4 shows that the adaptive attack has little impact on the final model accuracy though it is stealthier than the continuous attack. This is due to the similarity between the test gradient and the normal aggregated gradient, leading to inaccurate estimation by the attacker and a decrease in aggregation weight. Also, increasing detection frequency lowers poisoning frequency, so the poisoning impact will be gradually alleviated by normal training over time.

**Poisoning before the Algorithm Initialization.** The detection basis is that the variance of the malicious gradient is larger than the benign gradient. Malicious gradients, solved by the optimization problem, are more inconsistent. It is also theoretically proven that the optimization problem amplifies this variance. Thus, RECESS is unaffected by initial poisoned gradients as it does not change this basis. Besides, when the poisoned aggregated gradient was used to detect, most clients will be identified as malicious. This violates the assumption of Byzantine robustness that requires more than 51% of users to be benign. Hence, the defender can easily discern a potentially poisoned initial gradient based on the new abnormality definition of RECESS.

**Inconsistent attacks.** Attackers can also first pretend as benign client to increase the trust score, and then provide the poisoning gradients. However, RECESS detects such behavior inconsistencies over time. The trust scoring of RECESS also incorporates delayed penalties for discrepancies between a client's current and past behaviors (shown in Section 3.3). Additionally, three factors limit intermittent attacks' impact:

(a) FL's self-correcting property means inconsistent poisoning is much less impactful. Attackers would not take this approach in practice.

(b) In real settings, clients participate briefly, often just once. Attackers cannot afford to waste rounds acting benign before attacking.

(c) Defenses aim to accurately detect malicious updates for model convergence. Even if poisoning temporarily evaded detection, attack efficacy would diminish significantly, making it no longer a serious security concern.

## 5  Discussion

This work primarily focuses on untargeted model poisoning, along with previous SOTA works. We believe that model poisoning has a significant impact on real-world FL deployments for three reasons:

(a) Model poisoning through direct gradient manipulation poses a greater threat than backdoors.

(b) Backdoors are considered special cases of model poisoning, as malicious gradients can be obtained on poisoning datasets. However, it is challenging to solely imitate model poisoning gradients through training backdoors.

(c) Untargeted poisoning has a more severe impact on overall model performance degradation compared to backdoors in FL.

As RECESS detects inconsistencies between client behaviors, it could mitigate targeted attacks from clients with backdoored data by deploying additional strategies. Due to space limitation, we refer the readers to our technical report [20] for more details.

## 6  Conclusion

In this work, we proposed RECESS, a novel defense for FL against the latest model poisoning attacks. We shifted the classic reactive analysis to proactive detection and offered a more robust aggregation mechanism for the server. In the comprehensive evaluation, RECESS achieves better performance than SOTA defenses and solves the outlier detection problem that previous methods can not handle. We anticipate that our RECESS will provide a new research direction for poisoning attack defense and promote the application of highly robust FL in practice.

### Acknowledgments

We thank all the anonymous chairs and reviewers for their valuable guidance and constructive feedback. Hui Li is partially supported by the National Natural Science Foundation of China (61932015), the National Key Research and Development Program of China (2022YFB3104100), Shaanxi innovation team project (2018TD-007), and Higher Education Discipline Innovation 111 project (B16037). Wenjing Zhang and Xiaodong Lin are partially supported by Natural Sciences and Engineering Research Council of Canada (NSERC). Part of Haonan Yan's work is done when he visits the School of Computer Science at the University of Guelph.

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
