# Appendix of RECESS

## A  Additional Related Works

### A.1  Federated Learning

**FedAvg.**  FedAvg [6] is a commonly used aggregation rule for FL. The aggregation gradient is a weighted average of each client's upload gradient, and the weight is determined by the number of training data. However, the aggregation gradient, i.e., the global model, is vulnerable to poisoning attacks, resulting in a global model with poor prediction performance [9].

### A.2  Poisoning Attacks in FL

From the perspective of the attacker's goal, poisoning attacks are categorized as targeted and untargeted attacks. The targeted attack [20, 8] aims to mislead the global model to misclassify samples of one attacker-chosen class, does not affect other classes. The untargeted attack [21, 7, 22] aims to increase the testing error of the global model for all classes. These poisoning goals can be accomplished by different operations, which are mainly classified into two categories, data and model poisoning attacks. Data poisoning attacks [23, 24] contaminate the training dataset to mislead the training process of local models in FL. Model poisoning attacks [13, 12] directly poison the gradient updates to corrupt the server's aggregation.

### A.3  Byzantine-robust Aggregation Rules

**Krum.**  [9] propose a majority-based approach called Krum. Suppose there are $n$ clients and $c$ malicious clients, Krum selects one gradient that minimizes the sum of squared distances to $n - c - 2$ neighbors as the final aggregation result.

**Mkrum.**  [9] propose a variant of Krum called Mkrum. Mkrum iteratively uses the Krum function $m$ times to select the set of gradients without put-back, and the final aggregation gradient is the mean of the selection set. Note that Mkrum is Krum when $m = 1$, and Mkrum is FedAvg when $m = n$.

**Trmean.**  [10] improve the FedAvg and propose coordinate-wise Trmean. Trmean removes the smallest and largest $k$ gradients and averages the remaining gradient as the aggregation result.

**Median.**  [10] propose another coordinate-wise aggregation rule called Median. This rule uses the median value of each dimension value of all upload gradients as the final output.

**Bulyan.**  [7] combine the Krum and Trmean into a new robust aggregation rule called Bulyan. Bulyan first selects several gradients with Krum and then takes the trimmed mean value of the selected set as the final result.

### A.4  Other Defenses

**Gradient Clipping.**  The gradient clipping strategy directly normalizes the gradient, which severely slows down the training, while RECESS does not modify the gradient itself, we only consider the magnitude as one of the factors to measure abnormality. Furthermore, clipping is a one-size-fits-all approach that does not consider benign outliers, reducing the model generalization, and cannot effectively resist the latest poisoning attacks, while RECESS addresses these issues.

**FLTrust.**  FLTrust involves the server with a small dataset to participate in each iteration and generate a gradient benchmark in each iteration. Updates far from the benchmark will be reduced in aggregation weights. However, a small dataset is less representative, especially in Non-IID cases it is insufficient to represent all benign outliers where small datasets are less representative. Thus, FLTrust would discard benign outliers. In fact, using a static benchmark (e.g., median, FLTrust) to detect the malicious client will always implicate benign outliers. Nevertheless, RECESS improves this limitation through proactive detection and multi-round evaluation.

## B   Discussion

**Compliance.**    All clients just follow normal FL training without any extra rules to obey. Our threat model is consistent with previous works (FLTrust, DnC) and does not constrain or weaken the attacker. RECESS is running on the server side and considers various attack behaviors including cheating.

**Novelty.**    To the best of our knowledge, RECESS is the first defense consisting of proactive detection and robustness aggregation. Considering the lack of similar schemes in the literature, we demonstrate its effectiveness by comparing it with the SOTA defenses in detection and aggregation, respectively.

**Cost.**    As the first proactive detection, RECESS requires clients to cooperate, thus consuming certain iterations. For $r$ iterations of detection, the communication and computational cost are $r$ and $n * r$ respectively, where $n$ is the number of clients with customized test gradients. In practice, we find that $r$ is a small number thanks to our effective detection, e.g., $10$ is sufficient for most cases, which is negligible for the overall convergence (usually $500 - 1500$).

**Scalability.**    RECESS can adapt to different FL setting in which the server gets the local model updates and then aggregate them directly. This adaptation can be achieved by converting uploaded models into gradients, since sending model updates is considered equivalent to sending gradients.

Besides, it is an arms race to develop defences against new emerging attacks including backdoor. It can be considered from two perspectives: stealthiness and attack effect, so the corresponding defenses need to consider both detection and aggregation [25]. Thus, RECESS is a promising method of detecting new attacks including the backdoor since it has been improved on both proactive detection and robustness aggregation. For more stealthy attacks, RECESS can lengthen the detection window and also make the detection more sensitive to gradient changes by adjusting the magnitude of test gradients. For attacks with better effect, RECESS can cope with the latest attack, and the robustness aggregation can effectively limit the effect of malicious gradients on aggregation.