# OpenReview forum: "RECESS Vaccine for Federated Learning: Proactive Defense Against Model Poisoning Attacks"
_NeurIPS.cc/2023/Conference — NeurIPS 2023 poster_

### Official Review · Reviewer_nW5D · 2023-06-30

**Soundness:** 3 good
**Presentation:** 3 good
**Contribution:** 3 good
**Rating:** 5
**Confidence:** 5

**Summary:**

In this paper, the authors propose RECESS, which is a proactive defense against untargeted model poisoning attacks. Specifically, RECESS proactively detects malicious clients with test gradients and robustly aggregates gradient with a new trust scoring based mechanism. For the former, the key insight is that the update goals of malicious clients are different from benign clients and outliers. Therefore, the defender can amplify this difference by delicately constructing aggregation gradients sent to clients, and then more accurately detect malicious clients and distinguish benign outliers by the comparison between constructed gradients and the corresponding responses. For the latter, a new trust scoring based mechanism is introduced to estimate the score according to the user’s performance over multiple iterations. Experiments on four datasets and various settings including white/black-box, cross silo/device FL, etc are conducted. The authors also compare the proposed approach with five classic and two SOTA defenses.

**Strengths:**

The paper is well-motivated to solve two limitations in existing poisoning defense: (1) existing methods are susceptible to the optimization-based model poisoning attacks, leading to accuracy loss, and (2) existing methods cannot distinguish malicious gradient and benign outliers, reducing the generalization performance of the trained model. The proposed method is well-justified both theoretically and experimentally. Extensive evaluations on four datasets with various FL settings and attack settings, as well as the comparison with five classic and two SOTA defense are provided.

**Weaknesses:**

While this is a well-written and well-motivated submission, this reviewer has the following concerns.
       •	Limited capacity of defense. The observation that malicious clients aim to make the poisoned aggregation direction as far away from the aggregation gradient without poisoning as possible to maximize the poisoning effect only holds for untargeted model poisoning.
       •	Discussion of compliance missing. While the authors design the test gradients and send it to the clients for identifying the malicious clients, it is unclear how would the malicious attack comply with the designed the rule? Can malicious clients cheat on the test gradient?
       •	Technical details require additional clarification. (1) It seems for this reviewer, it is not clear how the “slight adjust the direction of g_t_i by heuristically selecting several elements” is done. What are those elements? (2) the models evaluated in the experiment are relatively simple and shallow. This raises the concern of scalability to deeper models. (3) Missing details for IID/Non-IID, cross-silo/cross-device configurations, such as how many clients or devices, how much data per client, how to generate the non-IID division.

**Questions:**

See weakness.

**Limitations:**

See weakness.

---

> ### Author Rebuttal · Authors · 2023-08-05
>
> We thank the reviewer for the thoughtful comments. We provide our responses and additional evaluations below to address the concerns.
>
> ---
>
> **Q1:** Limited capacity of defense.
>
> **R1:** We appreciate the comment that our method primarily handles untargeted model poisoning.
>
> This submission does focus on untargeted attacks, prevalent in prior arts. We believe untargeted attacks pose a more severe threat to FL, for three reasons:
>
> 1.   Model poisoning attacks directly manipulating gradients are more threatening than targeted backdoor and data poisoning in FL [8, 13], and highly relevant to production FL.
>
> 2.   Targeted backdoor and data poisoning can be seen as special cases of model poisoning as malicious gradients can be obtained from poisoning datasets [11]. However, it is non-trivial to imitate model poisoning gradients only via data poisoning.
>
> 3.   Untargeted poisoning degrades model performance more severely than targeted ones in FL [12-14].
>
> Nevertheless, by simply substituting robust aggregation for Byzantine-robust aggregation, our RECESS can effectively defend against other poisoning attacks. To demonstrate RECESS's versatility, we evaluated various attacks, including data poisoning (label flipping) and targeted backdoor (scaling attack).
>
> | Dataset  | Attacks                              | FedAvg     | FLTrust | DnC    | RECESS | RECESS with Median |
> | -------- | ------------------------------------ | ---------- | ------- | ------ | ------ | ------------------ |
> | MNIST    | No Attack (Model Accuracy)           | **0.9621** | 0.9584  | 0.9601 | 0.9598 | 0.9581             |
> |          | Label Flipping (Model Accuracy)      | 0.9018     | 0.9448  | 0.9225 | 0.9154 | **0.9548**         |
> |          | Scaling Attack (Attack Success Rate) | 1.0        | **0**   | 1.0    | 1.0    | **0**              |
> | CIFAR-10 | No  Attack                           | **0.6605** | 0.6341  | 0.6409 | 0.6554 | 0.6383             |
> |          | Label Flipping                       | 0.4145     | 0.5748  | 0.4284 | 0.4415 | **0.6148**         |
> |          | Scaling Attack                       | 1.0        | 0.0248  | 0.6412 | 1.0    | **0**              |
>
> The setting of the label flipping attack is the same as [13] and the scaling’s setting is consistent with [23]. The accuracy of the model using RECESS with Median under all attacks rivals conventional models (using FedAvg under no attack). This shows RECESS can effectively defend against diverse FL poisoning. While not demonstrated originally, we respectfully suggest our technique is capable of tackling these attacks straightforwardly.
>
> ------
>
> **Q2:** Discussion of compliance missing.
>
> **R2:** Thank you for the feedback to strengthen the compliance discussion. Attackers may attempt to evade detection, but we account for potential adaptive attack behaviors and have countermeasures, including increased detection frequency, heightened parameter sensitivity, and delayed reward. These can effectively restrict malicious clients and ensure proper FL system operation. The evaluation of adaptive attacks was covered comprehensively in **Section 4.4**.
>
> ---
>
> **Q3:** Additional technical details.
>
> **Q3.1:** Details of gradient modification.
>
> **R3.1:** Thank you for the feedback. Due to page limits, more specifics are in the appendix. As described in **Appendix E.3**, the modification allows arbitrary adjustments to gradient elements within a defined threshold. In experiments, we select and add noise to the first 10% of dimensions, iteratively adjusting until cosine similarity before/after meets the threshold.
>
> ---
>
> **Q3.2:** The concern of scalability to deeper models.
>
> **R3.2:** Thank you for raising this important point. For fair evaluation, we used the same base models as prior works. Nevertheless, our approach can extend to deeper networks. To demonstrate scalability, we conducted additional experiments on larger models including DNN, VGG, and ResNet. As shown in the table below, our method achieves consistent utility for these complex models.
>
> | Dataset  | Model    | Attacks             | FedAvg     | RECESS     |
> | -------- | -------- | ------------------- | ---------- | ---------- |
> | MNIST    | DNN      | No Attack           | **0.9487** | 0.9405     |
> |          |          | Optimization attack | 0.6873     | **0.9314** |
> | CIFAR-10 | ResNet20 | No Attack           | **0.8449** | 0.8217     |
> |          |          | Optimization attack | 0.1718     | **0.8173** |
> |          |          | AGR-tailored Attack | 0.1544     | **0.8014** |
> |          | VGG11    | No Attack           | **0.7515** | 0.7408     |
> |          |          | Optimization attack | 0.5032     | **0.7344** |
> |          |          | AGR-tailored Attack | 0.3834     | **0.7037** |
>
> These results highlight the general applicability of our framework across model depths.
>
> ---
>
> **Q3.3:** More details needed on IID/Non-IID and cross-device/cross-silo settings like number of clients, data per client, and how non-IID data is generated.
>
> **R3.3:** Similarly, we use the same settings as prior works for fair comparison:
>
> *   Number of clients/devices specified in **Table 1** for each dataset. For cross-device FL, the server aggregates from 10 random clients per round on CIFAR-10 and 60 random clients on FEMNIST, a common setting in prior work.
> *   CIFAR-10/MNIST: 50 clients with 1000 samples each. Purchase: 80 clients with 2000 samples each. FEMNIST uses a native non-IID partition of 3400 clients owning unique data.
>
> *   Except FEMNIST is natively non-IID, non-IID CIFAR-10 uses the standard approach from prior work. With *M* classes, clients split into *M* groups. Class *i* example assigned to group *i* with probability *q*, and to any group with probability *(1-q)/(M-1)*. *q=1/M* is IID, larger *q* means more non-IID.
>
> Please let us know if any additional clarification on the settings would be helpful. We are happy to provide more details on the configurations used.

---

> ### Author Response · Authors · 2023-08-21
> **Any additional questions?**
>
> Hi Reviewer nW5D #4,
>
>
> Thank you for taking the time to review our work.
>
> We wanted to kindly ask if our responses have sufficiently addressed the concerns you previously raised.
>
> Please let us know if you need any additional clarification or have any other questions. We are happy to provide more details if needed.
>
>
> Best regards,
>
> Authors of Submission7493

---

### Official Review · Reviewer_tdiM · 2023-07-03

**Soundness:** 3 good
**Presentation:** 3 good
**Contribution:** 3 good
**Rating:** 7
**Confidence:** 4

**Summary:**

This paper proposes a new defense against model poisoning attacks in FL. The idea is that the server sends some perturbed aggregation gradient to clients in some selected training rounds, and based on the responses, the server adjusts trust scores for the clients. Experimental results show the effectiveness of the proposed defense against state of the art attacks and outperforms state-of-the-art defenses.

**Strengths:**

+ Important and relevant research problem.

+ Well written paper.

+ Interesting and novel approach.

**Weaknesses:**

I didn't see severe weaknesses. I think the paper is above the bar.

I have three suggestions:

1. Since the paper talks about detection. I would suggest also comparing with detection methods, e.g., the following:

FLDetector: Defending federated learning against model poisoning attacks via detecting malicious clients. In KDD, 2022.

This detection method may also be adapted to assign trust scores for clients.

2. All the evaluated attacks are for compromised genuine clients. Recent attacks use fake clients, e.g., the following:

Mpaf: Model poisoning attacks to federated learning based on fake clients. In CVPR Workshop, 2022.

The paper may want to evaluate such attacks.

3. The paper seems to assume that the server sends aggregated gradient to the clients. This is different from standard FL. The paper can make this more clear. Algorithmically, it is equivalent to send the new global model to clients, but it's good to make this clear.


**Questions:**

See above.

**Limitations:**

See above.

---

> ### Author Rebuttal · Authors · 2023-08-05
>
> Thanks to the reviewers' insightful suggestions, which are helpful to strengthen the completeness of this work.
>
> ---
>
> I didn't see severe weaknesses. I think the paper is above the bar. I have three suggestions:
>
> **S1:** Since the paper talks about detection. I would suggest also comparing with detection methods, e.g., the following:
>
> FLDetector: Defending federated learning against model poisoning attacks via detecting malicious clients. In KDD, 2022.
>
> This detection method may also be adapted to assign trust scores for clients.
>
> **R1:** Yes, FLDetector also adopts a "trust scoring" approach to identify malicious clients. The key difference lies in how anomalies are detected. We actively send test queries and compare the updates returned from clients, while FLDetector makes predictions based on historical updates. We also evaluated FLDetector, with the following results:
>
> | Dataset                | Attacks              | FLDetector | RECESS |
> | ---------------------- | -------------------- | ---------- | ------ |
> | CIFAR-10 (IID)         | Optimization attack  | 0.6384     | **0.6390**|
> |                        | AGR-agnostic Min-Max | 0.5178     | **0.6286** |
> | CIFAR-10 (Non-IID 0.5) | Optimization attack  | 0.6041     | **0.6145** |
> |                        | AGR-agnostic Min-Max | 0.5514     | **0.5912** |
> | CIFAR-10 (Non-IID 0.8) | Optimization attack  | 0.3544     | **0.6018** |
> |                        | AGR-agnostic Min-Max | 0.0176     | **0.5841** |
> | FEMNIST                | Optimization attack  | 0.7217     | **0.7937** |
> |                        | AGR-agnostic Min-Max | 0.5616     | **0.7850** |
>
> In regular settings, the performance of FLDetector (Median) is comparable to RECESS, but when facing stronger attacks (AGR attack series) or higher Non-IID levels (>0.5 or on FEMNIST task), the model accuracy achieved by RECESS is significantly higher than FLDetector. This is because as the number of benign outliers increases significantly, the fluctuations in clients’ historical gradients become more intense, which greatly reduces the accuracy of FLDetector's predictions. In contrast, RECESS can effectively cope through dynamic testing. Therefore, the comparative experiments with FLDetector further demonstrate RECESS's advantages in tackling the challenging problem of distinguishing between benign outliers and malicious users.
>
> ---
>
> **S2:** All the evaluated attacks are for compromised genuine clients. Recent attacks use fake clients, e.g., the following:
>
> Mpaf: Model poisoning attacks to federated learning based on fake clients. In CVPR Workshop, 2022.
>
> The paper may want to evaluate such attacks.
>
> **R2:** Thank you for the reviewer's suggestion to evaluate this new attack type named MPAF.
>
> We thoroughly read this attack scheme. This attack injects fake clients rather than compromising victim clients, allowing a higher ratio of malicious users. In addition, MPAF uses a very simple attack method that merely attempts to drag the global model towards an attacker-chosen low-accuracy model, with a fixed local update goal and without optimization during poisoning. RECESS does not consider this type of poisoning attack, mainly because:
>
> 1.  The attacker's assumptions do not conform to real-world FL requirements. We have concerns about its practical feasibility due to the strong assumption of arbitrary fake client injection required. To the best of our knowledge, satisfying such assumptions is highly challenging in practice, as evidenced by the following literature:
>
>     *Shejwalkar, Virat, Amir Houmansadr, Peter Kairouz, and Daniel Ramage. "Back to the drawing board: A critical evaluation of poisoning attacks on production federated learning." In 2022 IEEE Symposium on Security and Privacy (SP), pp. 1354-1371. IEEE, 2022.*
>
> 2.  Even if these assumptions held, basic Byzantine-robust aggregation rules could readily defeat the attack. Our proposal can be enhanced by using a common Median or Krum instead of weighted averaging in RECESS to defend against MPAF.
>
> 3.  Although MPAF submits malicious gradients, it is essentially a data poisoning attack before the FL system starts. Existing detections during training, like FLDetector, also do not consider such attacks.
>
> While we appreciate the reviewer's perspective to include recent advancements, given our reservations on the practicality of this attack, we are worried that elaborating on it in depth might give readers the false impression that it poses a serious threat. If the reviewer still believes a comparison is needed, we would be happy to reconsider given further guidance. Otherwise, we hope to focus our manuscript on defending against attacks under more realistic assumptions that are standard in the field.
>
> ---
>
> **S3:** The paper seems to assume that the server sends aggregated gradient to the clients. This is different from standard FL. The paper can make this more clear. Algorithmically, it is equivalent to sending the new global model to clients, but it's good to make this clear.
>
> **R3:** Thank you for pointing this out. Yes, to actively test malicious clients, the server with RECESS deployed needs to send gradients rather than the model weight to clients, and these two approaches are equivalent algorithmically. We will clarify this in a future version.
>
> Please let us know if we can provide any clarification or justify our position further. Overall, we sincerely thank the reviewer for the time and help in strengthening our work.

---

> ### Author Response · Authors · 2023-08-21
> **Thank you for your comments.**
>
> Hi Reviewer tdiM #3,
>
> Thank you for your thoughtful comments on our paper. We appreciate you taking the time to provide feedback, as it will help strengthen our work.
>
> Please let us know if you would like us to clarify or expand on any part of the paper before the end of the reviewer-author discussion period. We look forward to continuing the conversation and thank you again for your review.
>
> Best regards,
>
> Authors of Submission7493

---

### Official Review · Reviewer_MKss · 2023-07-07

**Soundness:** 2 fair
**Presentation:** 3 good
**Contribution:** 2 fair
**Rating:** 5
**Confidence:** 4

**Summary:**

The paper presents RECESS, a backdoor defense method in federated learning. The central server keeps querying each client and tries to estimate the trust score of each client to determine if it is malicious or not. The underline intuition of the anomaly detection method is that the malicious client will push the gradient far away from benign directions.


**Strengths:**

The paper is clearly written with clear labels and easy-to-follow logic.

The method is evaluated on several datasets and federated learning settings.

The paper considers the adaptive settings to evaluate its strengths.


**Weaknesses:**

The paper does not validate its assumption. The goal of the adversary in poisoning attacks is to mislead the output with the trigger, and there is no need to manipulate the malicious gradient so that it is far away from benign ones. In fact, many works have demonstrated that some "unintended backdoors" only have benign samples, and there are no real malicious triggers. The adversary can later generate triggers based on the trained model (trained on only benign data). As such, I think the paper needs to validate its own assumption or intuition before applying it to practice, especially considering that this may be wrong.

I do not see how the method overcomes the limitations of existing methods. The paper mentions that existing work has false positives, i.e., identifying benign ones as malicious. In a typical federated setting with non-IID data, it is common that some clients will have significantly different gradient optimization directions, which can be far away from others and invalidate the detection assumption of this method. How does this method prevent such false positives from happening?

The evaluation does not consider methods with theoretical guarantees or stronger adaptive attackers. For example, is it possible to give benign responses to only test gradients while sending malicious gradients to the server?


**Questions:**

See above.


**Limitations:**

The paper does not discuss its limitations.

---

> ### Author Rebuttal · Authors · 2023-08-05
>
> Thanks for the valuable feedback. Please find our response below.
> ___
>
> **Q1:** The paper does not validate its assumption. The goal of the adversary in poisoning attacks is to mislead the output with the trigger, and there is no need to manipulate the malicious gradient so that it is far away from benign ones. In fact, many works have demonstrated that some "unintended backdoors" only have benign samples, and there are no real malicious triggers. The adversary can later generate triggers based on the trained model (trained on only benign data). As such, I think the paper needs to validate its own assumption or intuition before applying it to practice, especially considering that this may be wrong.
>
> **R1:** Thank you for the opportunity to clarify the attack assumptions. We agree it is important to articulate these upfront in the submission.
>
> Our assumed model poisoning adversary directly manipulates gradients to cause untargeted harm to server aggregation, representing a new type of attack on FL proposed recently. This differs from traditional data poisoning or targeted backdoor attacks. As the reviewer points out, we should state this distinction more clearly in the main sections, not just the Appendix. Model poisoning has distinct motivations and methods compared to data attacks. As cited [8,13], this threat model has been adopted by related works and poses severe risks in real-world FL:
>
> 1.   Model poisoning via direct gradient manipulation is more threatening than targeted backdoor attacks [8, 13], and highly relevant to production FL.
> 2.   Targeted backdoor attacks are considered special cases of model poisoning as malicious gradients can be obtained on poisoning datasets [11]. However, it is non-trivial to imitate model poisoning gradients only via data poisoning.
>
> 3.   Untargeted poisoning is a more severe threat to model performance than targeted ones [12-14].
>
> We apologize for the lack of clarity on threat model assumptions and will move the description in “Appendix A.1 & F.2” to **Related Work**.
>
> Although RECESS focuses on model poisoning, we show by replacing weighted averaging with Median aggregation, RECESS can mitigate the scaling attack, a representative backdoor.
>
> | Dataset  |               Attacks                |   FedAvg   | FLTrust |  DnC   | RECESS | RECESS with Median |
> | :------: | :----------------------------------: | :--------: | :-----: | :----: | :----: | :----------------: |
> |  MNIST   |   No Attack (Model Test Accuracy)    | **0.9621** | 0.9584  | 0.9601 | 0.9598 |       0.9581       |
> |          |   Scaling Attack (Model Accuracy)    | **0.9542** | 0.9428  | 0.9518 | 0.9539 |       0.9519       |
> |          | Scaling Attack (Attack Success Rate) |     1      | **0.0** |   1    |   1    |      **0.0**       |
> | CIFAR-10 |   No Attack (Model Test Accuracy)    | **0.6605** | 0.6341  | 0.6409 | 0.6554 |       0.6383       |
> |          |   Scaling Attack (Model Accuracy)    | **0.6329** | 0.6168  | 0.6123 | 0.6314 |       0.6228       |
> |          | Scaling Attack (Attack Success Rate) |     1      | 0.0248  | 0.6412 |   1    |     **0.0127**     |
>
> Additional results demonstrate that RECESS with Median achieves comparable accuracy to the normal model using FedAvg under no attack and resists the backdoor effectively.
>
> ---
>
> **Q2**:  I do not see how the method overcomes the limitations of existing methods. The paper mentions that existing work has false positives, i.e., identifying benign ones as malicious. In a typical federated setting with non-IID data, it is common that some clients will have significantly different gradient optimization directions, which can be far away from others and invalidate the detection assumption of this method. How does this method prevent such false positives from happening?
>
> **R2:** We introduce a new detection approach, proactive probing, to address the intractable problem of benign outlier identification that passive statistics-based defenses struggle with, especially when the degree of non-IID is large, this problem is more difficult to solve.
>
> Instead of previously taking a single fixed benchmark to determine anomalies, we make judgments based on the consistency of the client's behavior over time to avoid detection errors. Specifically,
>
> -   The key insight is that malicious and benign clients (including benign outliers) have fundamentally different update goals. By carefully constructing the aggregation gradients sent to clients, we can amplify this difference and more accurately detect malicious behaviors while distinguishing benign outliers.
> -   Additionally, we propose a new trust scoring-based robust aggregation mechanism that estimates scores based on long-term user performance across iterations, rather than scoring each round independently. This improves fault tolerance and robustness.
>
> Together, the novel concepts of proactive probing and robust trust scoring allow RECESS to outperform SOTA defenses against the latest poisoning attacks, while accurately identifying benign outlier gradients.
>
> ---
>
> **Q3:** The evaluation does not consider methods with theoretical guarantees or stronger adaptive attackers. For example, is it possible to give benign responses to only test gradients while sending malicious gradients to the server?
>
> **R3:** We compare five classic robust rules and two SOTA works, all with solid theoretical guarantees. We also analyze RECESS under black box and white box settings and provide theoretical guarantees, as shown in **Proposition 1 in Section 3.4 Effectiveness Analysis** in lines 196-205 of page 6. The proof can be found in **Section D Proof of Proposition 1** in lines 447-467 of page 13.
>
> Stronger adaptive attackers will struggle to accurately determine if crafted gradients are tests, limiting poisoning impact under our defense. We refer the reviewers to the evaluation of adaptive attacks in **Section 4.4 Adaptive Attacks** in lines 287-312 of page 9 for more details.

---

> > ### Comment · Reviewer_MKss · 2023-08-17
> >
> > Thank you for the clarification, especially the threat model part. Now it is clear to me, and I have raised the score to borderline accept with some concerns about the significance of the work (Q2).

---

> > > ### Author Response · Authors · 2023-08-17
> > > **Thank you and more clarification**
> > >
> > > Thank you for your feedback and recognition of our previous response. We attempt to further address your concerns about the significance of the work (Q2).
> > >
> > >
> > > We first describe **two limitations we have investigated in the current model poisoning detection field** to better motivate the significance of our work and clarify the key contributions.
> > >
> > > -   Existing defense mechanisms have been shown ineffective against the latest optimization-based model poisoning attacks like the AGR attack series.
> > >
> > > -   As the reviewer also mentioned, current defenses suffer from high false positive rates, i.e. identifying benign outlier clients as malicious, especially in non-IID settings where benign outliers are common yet easily misidentified as malicious.
> > >
> > >
> > >
> > > **Why are existing defense mechanisms ineffective?**  Mainly because of their reliance on passive statistical analysis of uploaded information from clients, whether via Byzantine-robust aggregation (Krum, Trmean, etc.), uploaded gradient analysis (FLTrust, DnC), or prediction from historical updates (FLDetector). By adopting fixed benchmarks or statistical majorities, these schemes inevitably sacrifice benign outliers, which is also the most challenging issue in anomaly detection.
> > >
> > >
> > >
> > > To further showcase significance, we expand the discussion on **how our work addresses the limitations of prior techniques**. To resolve the aforementioned limitations, we take a brand-new, never seen before approach suitable for FL - the server proactively probes and discovers more behavioral characteristics of benign and malicious clients, instead of passively analyzing limited information from clients.
> > >
> > > Our **key observation** is that while some clients have different gradient directions, the core difference is: benign clients including outliers optimize the received aggregation result to the distribution of their local dataset to minimize the loss value. In contrast, malicious clients aim to maximize the poisoning effect by pushing the poisoned aggregation direction as far away as possible from the non-poisoned aggregation direction. In other words, benign gradients explicitly point towards their local distribution which is directional and relatively more stable, while malicious gradients change inconsistently and dramatically as they are influenced by benign ones.
> > >
> > > Leveraging this insight, we amplify this difference by delicately constructing aggregation gradients sent to clients, and then more accurately detect malicious clients and distinguish benign outliers by the comparison between constructed gradients and the corresponding responses. Additionally, to increase tolerance and robustness, we do not perform single-round aggregation as in previous work (e.g. FLTrust), because no mechanism can guarantee always detection correctly. Instead, we design a trust score based on the long-term behavior over multiple rounds for each client, which means even if we misdetect some rounds, clients that generally behave well will not be excluded as in previous schemes.
> > >
> > >
> > >
> > >
> > >
> > >
> > >
> > > **In summary, the significance of this work is:**
> > >
> > > Unlike previous anomaly detection-based defenses, this work proposes a novel proactive defense, tailored for FL, against the latest model poisoning attacks. We shift from the classic reactive analysis to proactive detection and offer a more robust aggregation mechanism for the server. As evidence, we directly compare to recent SOTA schemes, demonstrating improved defense over current methods on typical datasets under various FL settings, better positioning how our method departs from and advances beyond current literature. This representing a step forward for model poisoning attack detection. Specifically, this work makes significant contributions in :
> > >
> > > -   Discovering and solving the intractable benign outlier identification, improving generalization. To our knowledge, the proposed proactive detection is the first to address the benign outlier issue, effectively differentiating them from malicious data and reducing false positives.
> > >
> > > -   High detection fault tolerance. The benign clients’ contributions are guaranteed by the trust scoring mechanism based on long-term multi-round behavior even with detection errors. This greatly improves fault tolerance and robustness of detection. Meanwhile, adaptive attacks can be effectively handled only by adjusting the detection frequency or the parameter sensitivity, etc.
> > >
> > > -   Practical wide applicability. We consider more practical real-world FL products than previous discussions, particularly the non-IID setting and cross-silo/device scenarios. Our method provides highly robust FL training, promoting the generalization and utility of FL in practice.
> > >
> > >
> > >
> > > Please let us know if you have any additional questions or require further clarification. We are happy to address them before the discussion ends.

---

> > > ### Author Response · Authors · 2023-08-21
> > > **Is further clarification required?**
> > >
> > > Hi Reviewer MKss #2,
> > >
> > > We greatly appreciate you taking the time to provide thoughtful feedback on our initial submission and response.
> > >
> > > We aimed to thoroughly address your concerns regarding the significance of the work and wanted to check if our explanations and additions were sufficiently clear in demonstrating the novelty and importance of this work.
> > >
> > > Please let us know if you have any other questions or need any clarification on these points before the reviewer-author discussion ends. We are happy to discuss this further if needed.
> > >
> > > Thank you again for your time and consideration.
> > >
> > > Best regards,
> > >
> > > Authors of Submission7493

---

### Official Review · Reviewer_St5y · 2023-07-24

**Soundness:** 3 good
**Presentation:** 2 fair
**Contribution:** 2 fair
**Rating:** 5
**Confidence:** 4

**Summary:**

To defend the untargeted poisoning attack on the federated learning, the authors proposed a new defense called as RECESS, which exploits the outlier detection to analyze the gradients returned from clients. Once the gradient from one client is judged as outlier, this client will be considered as malicious client. The outlier detection is based on the intuition, i.e., for the benign samples even including the outliers, their updated gradients always point to the direction leading to a (local) minimum loss value, while the gradients of poisoned data will point to the reversed direction.

**Strengths:**

Similar as previous refs [11, 12] as shown in the paper, this work aims to alarm a malicious gradient from a group of gradients updated from clients in the federated learning. The paper is well-written and can be easily understood.

**Weaknesses:**

The main weakness of this work is due to its trivial contribution compared with the STOA [11, 12]. The main intuition for outlier detection, as mentioned in ‘Summary’, has been considered. Compared with STOA, the new metric ‘trust scoring’ does not improve too much as shown in Table2.

**Questions:**

For the trust score, is it possible for attackers to first pretend as benign client to increase the trust score, and then provide the poisoning gradient? How should the defender limit this kind of attack? Could the attacker perform the adaptive attack to poison the global model?

**Limitations:**

One limitation of this work is that it only considers the non-target attack, which is less stealthy than target backdoor attack. The main reason is that the non-target attack’s goal is to affect the normal performance of the model, which can be easily recognized by the trainer. I recommend the authors mentioned this limitation in the paper. For more details on the difference between target and non-target attack, please check this survey.

E. Cinà, K. Grosse, A. Demontis, S. Vascon, W. Zellinger, B. A. Moser, A. Oprea, B. Biggio, M. Pelillo, and F. Roli. Wild patterns reloaded: A survey of machine learning security against training data poisoning. ACM Comp. Surveys, 2023.

Guo, Wei, Benedetta Tondi, and Mauro Barni. "An overview of backdoor attacks against deep neural networks and possible defences." IEEE Open Journal of Signal Processing (2022).

---

> ### Author Rebuttal · Authors · 2023-08-05
>
> Thank you very much for the valuable comments. Please find our responses below.
>
> ---
>
> **Q1:** The main weakness of this work is its trivial contribution compared to STOA [11, 12]. The intuition for outlier detection mentioned in the Summary has been considered. Compared to STOA, the new 'trust scoring' metric does not improve much as shown in Table 2.
>
> **R1:** Our contribution is non-trivial compared with previous SOTA works.
>
> We agree existing defenses rely on passive outlier detection, while our RECESS takes a novel proactive testing approach to identify poisoning. As noted, benign outliers and malicious gradients are numerically similar, making the distinction intractable with only existing passive analysis. Our key intuition is benign and malicious behaviors diverge when aggregated gradients are carefully modified for testing. This active detection provides a fresh perspective unseen in current defenses. Leveraging this, RECESS significantly outperforms existing methods against latest model poisoning attacks. The newly proposed "Abnormality" metric (Equation 2), instead of the conventional ‘trust scoring’, provides fundamental advances over previous static defenses. We believe these concepts make our work a non-trivial improvement in poisoning detection.
>
> To clarify, Table 2 shows RECESS achieves comparable performance to normal FedAvg under no attacks, and outperforms existing defenses on various tasks under typical settings. For example, under AGR-tailored attack, RECESS-trained model accuracy on FEMNIST is up to 44.84% higher than DnC, and 14.07% higher than FLTrust on CIFAR-10. We further evaluated more challenging settings favoring the attacker, like increasing non-IID (Figure 5), more malicious clients (Figure 6), and stronger threat models. In all scenarios, RECESS's advantages over current defenses become more significant. As defenses face sophisticated practical settings and stronger attacks, RECESS demonstrates greater robustness compared to SOTA approaches.
>
> ---
>
> **Q2:** For trust score, can attackers first pretend to be benign to increase it, then poison gradients? How to limit this attack? Could attackers perform adaptive attacks to poison the global model?
>
> **R2:** We appreciate the concern about inconsistent attacks. However, RECESS detects such behavior inconsistencies over time. The trust scoring of RECESS also incorporates delayed penalties for discrepancies between a client's current and past behaviors (lines 180-184). Additionally, three factors limit intermittent attacks' impact:
>
> 1.  FL's self-correcting property means inconsistent poisoning is much less impactful. Attackers would not take this approach in practice.
> 2.  In real settings, clients participate briefly, often just once. Attackers cannot afford to waste rounds acting benign before attacking.
> 3.  Defenses aim to accurately detect malicious updates for model convergence. Even if poisoning temporarily evaded detection, attack efficacy would diminish significantly, making it no longer a serious security concern.
>
> We evaluate more adaptive attacks in **Section 4.4**, including evasion and poisoning before initialization. Results show RECESS provides robustness against these strong adaptive attacks.
>
> ---
> **Recommendation:** One limitation is this work only considers non-target attacks, which are less stealthy than target backdoor attacks. The reason is that non-target attacks aim to affect model performance, which can be recognized by the trainer. I recommend mentioning this limitation.
>
> **Response:** We appreciate the constructive recommendation to evaluate targeted backdoor attacks. This submission focuses primarily on untargeted model poisoning, along with previous SOTA works. We believe model poisoning is particularly impactful to real-world FL deployments for three reasons:
>
> 1.   Model poisoning via direct gradient manipulation is more threatening than backdoors in FL [8,13].
> 2.   Backdoors are considered special cases of model poisoning as malicious gradients can be obtained on poisoning datasets [11]. However, it is non-trivial to imitate model poisoning gradients only via training backdoors.
> 3.   Untargeted poisoning degrades overall model performance more severely than backdoors in FL [12-14].
>
> Since RECESS detects inconsistency between client behaviors, it could mitigate targeted attacks from clients with backdoored data by replacing weighted averaging with a Byzantine-robust aggregation like Median. We added experiments showing RECESS with Median effectively defends against the scaling attack, a representative backdoor. RECESS with Median achieves comparable accuracy to the normal model under no attack and resists the backdoor.
>
> | Dataset  |               Attacks                |   FedAvg   | FLTrust |  DnC   | RECESS | RECESS with Median |
> | :------: | :----------------------------------: | :--------: | :-----: | :----: | :----: | :----------------: |
> |  MNIST   |   No Attack (Model Test Accuracy)    | **0.9621** | 0.9584  | 0.9601 | 0.9598 |       0.9581       |
> |          |   Scaling Attack (Model Accuracy)    | **0.9542** | 0.9428  | 0.9518 | 0.9539 |       0.9519       |
> |          | Scaling Attack (Attack Success Rate) |     1      | **0.0** |   1    |   1    |      **0.0**       |
> | CIFAR-10 |   No Attack (Model Test Accuracy)    | **0.6605** | 0.6341  | 0.6409 | 0.6554 |       0.6383       |
> |          |   Scaling Attack (Model Accuracy)    | **0.6329** | 0.6168  | 0.6123 | 0.6314 |       0.6228       |
> |          | Scaling Attack (Attack Success Rate) |     1      | 0.0248  | 0.6412 |   1    |     **0.0127**     |
>
> We also reviewed the suggested two surveys and will incorporate more comprehensive evaluations of targeted threats in future work to further highlight RECESS's capabilities. We appreciate this insightful discussion to strengthen and broaden the impact of our approach. Please let us know if you have any other suggestions to improve the coverage of additional attack types.

---

> ### Author Response · Authors · 2023-08-21
> **Has the question been resolved?**
>
> Hi Reviewer St5y #1,
>
> We greatly appreciate you taking the time to thoroughly review our work and provide constructive comments to improve the manuscript.
>
> We hope that our detailed responses in the rebuttal have sufficiently addressed your concerns regarding the contributions of our approach (which differs from the SOTA schemes of passive analysis by providing proactive detection), consideration of adaptive attacks (which we have already analyzed in the paper), and need for supplementary experiments (demonstrating that our method can resist more stealthy targeted backdoor attacks).
>
> Please let us know if our explanations and additional details have satisfactorily clarified these points or if you have any other remaining questions. We are happy to provide further information before the discussion period ends to fully address all aspects of your valuable feedback.
>
> Thank you again for your time and thoughtful consideration of our work.
>
> Best regards,
>
> Authors of Submission7493

---

### Decision · Program_Chairs · 2023-09-21

**Decision:**

Accept (poster)

**Comment:**

The work proposes a novel defense against model poisoning attacks to federated learning. The rebuttal addressed most of the concerns raised by the reviewers, and the authors are invited to modify the camera-ready accordingly. They should also improve the discussion on the limitations of the present work.